# Answering Questions Over Knowledge Graphs Using Logic Programming Along with Language Models

## Abstract

Question Answering over Knowledge Graphs (KGQA) is the task of answering natural language questions over a knowledge graph (KG). This task requires a model to reason over multiple edges of the KG to reach the right answer. In this work, we present a method to equip large language models (LLMs) with classic logical programming languages to provide an explainable solution to the problem. Our goal is to extract the representation of the question in the form of a Prolog query, which can then be used to answer the query programmatically. To demonstrate the effectiveness of this approach, we use the MetaQA dataset and show that our method finds the correct answer entities for all the questions in the test dataset.

## 1 Introduction

Logic programming is one of the most efficient approaches to finding the exact solution to a query over a knowledge graph since it generates an exact solution to a question. The main problem is that it is not easy to ask the right query given a natural language question. In this work we argue that for knowledge based question answering, we can separate the task of representing questions and memorizing factual knowledge. That is, instead of building models that embed entities of our knowledge graph, we can train the language model to learn the representation of the question in an intermediate query form and use a logical programming language to store our knowledge and reason over it based on the query. In this paper we show that its possible to equip language models with this powerful tool. We pick the task of multi-hop question answering over MetaQA dataset since the dataset provides the inference path for the questions and makes it easier to annotate each question with the corresponding logical form. This dataset contains approximately 300K training examples in which the answer entity is 1,2 or 3 hops away from the question entity in the knowledge graph. We build a knowledge base over the MetaQA dataset's knowledge graph using Prolog and fine-tune a sequence to sequence transformer Vaswani et al. (2017) to represent natural language questions to queries in logical form. Then, we use Prolog to answer the questions. We show that the model can answer all questions in the test dataset with 100% accuracy with less than 1% of the training data as compared to prior approaches which use full training dataset or a significant portion of it. Figure 1 shows the overall pipeline of our approach.

## 2 Related Work

A variety of approaches have been taken to address the problem of multi-hop question answering. A number of prior works have used graph embedding models to encode entities and relations in a knowledge graph and then score the triples in a KG and construct a scoring function so that the score for a correct triple is higher than the score of an incorrect one (Nickel et al., 2011; Yang et al., 2014a; Balazevic et al., 2019; Dettmers et al., 2017; Vashishth et al., 2019). Others have approached the problem by constructing a function that maps the question embedding along with an embedding of the graph or a subgraph around the question entity to the answer entity's embedding in knowledge graph (Sun et al., 2018; Saxena et al., 2020; Sun et al., 2019; He et al., 2021). Still others adapt a slightly different method, training a teacher model to learn intermediate signals and a student model to answer the questions (He et al., 2021). Finally, most relevant to our work, (Yang et al., 2014b) and (Yang et al., 2015) try to learn the logical form of the natural language questions by building a

semantic embedding space. However, our work differs from theirs in that we use LLMs to represent the question in logical form instead of manually building a semantic mapping space. The present work is thus the first to use large language models to represent questions in logical form and equip LLMs with logical programming tools to answer questions.

## 3 APPROACH

### 3.1 QUESTION TO LOGICAL FORM

In order to collect the dataset for fine-tuning a sequence-to-sequence transformer model, we use the multi-hop path given by the MetaQA dataset to get from question entity to the answer entity. We then map this path to the relations available in knowledge base and annotated each question with its corresponding predicates. In order for the model to only focus on the representation of the question itself, we replace the question entity with a string *ENT*. As an example, for the question *"the movies directed by [ENT] were written by who?"* we create the Prolog query **directed_by_reverse(ENT, X), written_by(X, Y)**. We randomly sample and annotate 100, 250, 500 and 1000 samples from the total 329,282 training examples and call those datasets *s100, s250, s500 and s1000* respectively. We describe the details of annotation in appendix C. We use a T5-small sequence-to-sequence transformer model to translate the question to an intermediate query representation. Finally, we train the model on each of the annotated training sets for 5000 training steps and chose the best model according to the exact match score on development dataset. We use the AdamW optimizer with an initial learning rate of 5e-5 and a linear learning rate scheduler, and train the model using a batch size of 8 on a single A100 GPU.

### 3.2 QUESTION ANSWERING

Figure 1 depicts the overall pipeline of our proposed model. We first transform each triple in the knowledge base of MetaQA to a first order logic predicate. As an example, for the triple *(Innocence, written_by, Hilary Brougher)* we construct the predicate **written_by(Innocence, Hilary Brougher)**. In order to capture reverse relations, for each of the 9 relations in MetaQA dataset we also create the reverse relation resulting in 18 total relations. Therefore, for the previous triple we also construct the triple **written_by_reverse(Hilary Brougher, Innocence)**. For any given question, we generate its logical form using the transformer model and then replace the *ENT* token with the entity ID from the knowledge graph to form the final Prolog query. Finally, we execute the query and retrieve the answers and the logical path from question entity to the answer entities.

## 4 EVALUATION AND RESULTS

MetaQA questions mostly come with multiple answers. Prior methods have used hit@1 as a metric to measure the performance of their model. This means that they measure if the highest ranked entity given by their model exists in the answer set. Our approach produces the exact solution path inside the knowledge graph and consequently it outputs all of the solutions (as depicted in Figure 1). For the sake of comparison, we also measure the hit@1 metric for our model over multi hop test datasets. In other words, the hit@1 score of our model is equivalent to the accuracy of finding all the entities in the answer set because our approach finds all possible answers or none of them. Table 1 compares our method with prior work. We report the scores for the model with best average score over 5 sampling and annotation iterations. We conclude that our model is capable of correctly answering all questions in the test dataset with only 1000 annotated examples.

## 5 CONCLUSION

We argue that the task of knowledge base question answering can be broken down into two components. First, representing the question and second, a logical reasoning engine that both holds the knowledge and performs the logical computation. We show the possibility of this approach by utilizing a small language model and Prolog as the logical programming language and testing it on the MetaQA dataset. Our method completely solves the knowledge based question answering challenge on this dataset.

URM STATEMENT

The authors acknowledge that at least one key author of this work meets the URM criteria of ICLR 2023 Tiny Papers Track.

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

A   PIPELINE OF THE MODEL AND COMPARISON TABLE

In this section we bring an overall view of the whole model's pipeline and also the table for comparing our results with prior work.

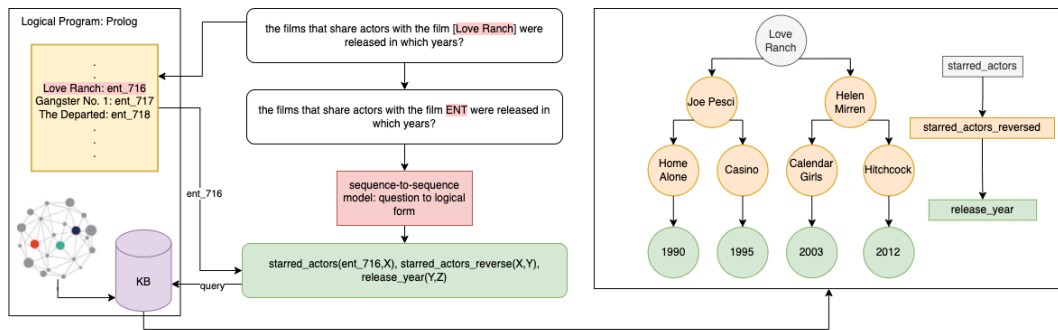

Figure 1: The complete inference pipeline of our proposed method. Note that the inference tree on the right side is a subset of the answer drawn here to clarify the schema of the model's output.

Table 1: Comparison of hit@1 score of previous methods compared to our method over multi-hop test datasets. The scores for the best model among 5 iterations of sampling is reported for our proposed method.

| Models | MetaQA-1hop | MetaQA-2hop | MetaQA-3hop |
|---|---|---|---|
| GraftNet (Sun et al., 2018) | 97.0 | 94.8 | 77.7 |
| PullNet (Sun et al., 2019) | 97.0 | 99.9 | 91.4 |
| EmbedKGQA (Saxena et al., 2020) | 97.5 | 98.8 | 94.8 |
| NSM (He et al., 2021) | 97.1 | 99.9 | 98.9 |
| T5-small+250 samples | 98.67 | 97.77 | **100.0** |
| T5-small+500 samples | **100.0** | 99.33 | **100.0** |
| T5-small+1000 samples | **100.0** | **100.0** | **100.0** |

## B   ROBUSTNESS OF THE METHOD

In order to get robust results we repeat the process of sampling training data and annotating it 5 times and each time we sampled 100, 250, 500 and 1000 samples. The sampling process was straightforward; each time we sample randomly and equally from each of the 1-hop, 2-hop and 3-hop training datasets. We also annotated 3000 samples from the validation and test set of the MetaQA dataset. Figure 2 shows the variance of performance on each of these datasets. Since the variance was high on the test set with 100 samples we only reported s250, s500 and s1000 in table 1. According to these results the 3-hop test set's representation is the easiest to learn since it doesn't come with many variations of natural language to describe. On the other hand the 2-hop dataset is the hardest to learn.

However, all of these samples are collected randomly. But with a manual and careful sample collection, we can see that even 500 samples are enough to learn the whole dynamics of this dataset and learn to represent questions in logical form.

## C   QUESTION TO LOGICAL FORM ANNOTATION

Each question in MetaQA dataset comes with the inference path inside the knowlege graph. For example, for the 2-hop question *"the movies written by [Hilary Brougher] were directed by who?"* there exists an inference path of *writer_movie_director* which shows the sequence of relations we need to traverse in the graph to reach the answer entity from the question entity *Hilary Brougher*. We use this inference path and annotate the question with the correct prolog query. To do so, we first break down the inference path into pairs. For the example above we would get *writer_movie* and *movie_director* pairs. Then we map each of the pairs to their corresponding predicate. Table C provides a list of all mappings that are available in the dataset. As mentioned in 3 we also replace

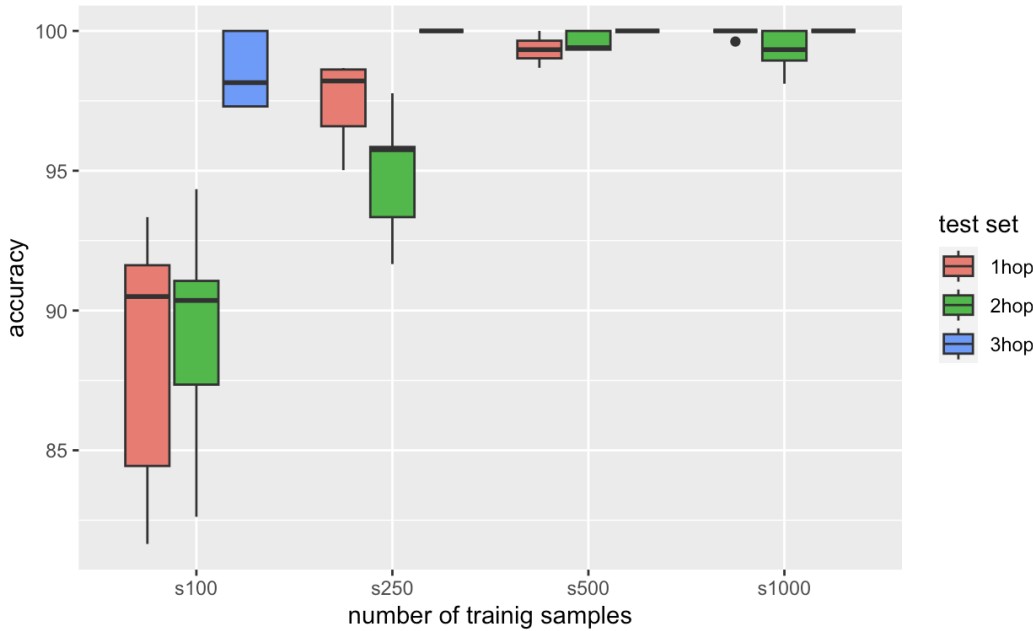

Figure 2: The variance of accuracy of each model based on the number of training examples available in each training dataset

Table 2: Mapping between different inference pairs and Prolog predicates

| Inference Pair | Predicate |
|---|---|
| actor_movie | starred_actors_reverse |
| director_movie | directed_by_reverse |
| movie_actor | starred_actors |
| movie_director | directed_by |
| movie_genre | has_genre |
| movie_imdbrating | has_imdb_rating |
| movie_imdbvotes | has_imdb_votes |
| movie_language | in_language |
| movie_tags | has_tags |
| movie_writer | written_by |
| movie_year | release_year |
| tag_movie | has_tags_reverse |
| writer_movie | written_by_reverse |

the question entity with token *ENT*. Afterwards, we start from the question entity and construct the logical form and replace the unknown entities in the path with X, Y and Z variables as needed. For example for the question above we construct the 2-hop logical form *written_by_reverse(ENT, X), directed_by(X, Y)*.

