# OpenReview forum: "Answering Questions Over Knowledge Graphs Using Logic Programming Along with Language Models"
_ICLR.cc/2023/TinyPapers — Submitted to Tiny Papers @ ICLR 2023_

### Official Review · Reviewer_5Ud5 · 2023-03-19

**Confidence:** 5

**Summary Of Contributions:**

KGQA task using T5-small with classic logical programming languages is presented.  The goal is to extract the representation of the question in the form of a Prolog query, which can then be used to answer the query programmatically. Using the MetaQA dataset, the presented method reaches 100% accuracy on the test dataset.

**Rating:**

Great Start (GS): a submission which meets some of the reviewing criteria but has room for improvement

**Strengths And Weaknesses:**

While the problem has been defined well, with some comparison to the literature, there are some very strong issues:
- 100% test accuracy is extremely unlikely for machine learning tasks unless the test data has been accidentally used to train the model.
- The authors must provide strong arguments to accept 100% test accuracy, which is never heard of.
- T5-small is used as the only large LM. Wonder how the other models will perform?
- The result table and Figure, which are required to follow the main text, are part of the body, NOT the appendix. As such, the page limit is not met. Also, Appendix C is the only referenced appendix. So how does the other relate?
- Experimental setup and hyperparameters are not provided. Hence, reproducibility is hard. Especially the claimed 100% accuracy.


**Suggested Changes:**

- The paper needs to be rewritten to fit the 2-page requirement while still using Appendix as additional material, NOT where the results are included.
- Include publications published in the last two years as part of the literature review.
- Explain 100% accuracy with a clear experimental setup for reproducibility.

This publication is a first step. But the authors need to clarify their method and results. Also, meet the basic requirements of a "Tiny" paper.

---

### Comment · Area_Chair_jCff · 2023-06-04
**No Revision is done**

This work has not been revised, and is not deanonymized

---

### Meta-Review · Area_Chair_jCff · 2023-04-06

**Recommendation:** Invite to revise
**Confidence:** 4

**Metareview:**

T5-small with classic logical programming languages, where the aim is to extract the representation of the question in the form of a Prolog query, which can then be used to answer the query programmatically. The authors achieve 100% accuracy on the test dataset.

The problem is defined well, but 100% test accuracy needs explanation. There must be clear explanations and arguments for accepting 100% test accuracy. The appendix should be used for additional/supportive material, not for results.

The publication as is doesn’t meet the basic requirements of the “Tiny” paper. The experimental setup needs to be clear for reproducibility.


**Summary:**

T5-small with classic logical programming languages, where the aim is to extract the representation of the question in the form of a Prolog query, which can then be used to answer the query programmatically. The authors achieve 100% accuracy on the test dataset.

**Comments And Feedback To The Authors:**

If you use a train-test split hold-out method for your evaluations, please run the experiment multiple times. Ensure the split is different each time. Make sure test data is not accidentally mixed with training data. Present the average score. Or consider k-fold cross-validation.

**Reason For Not Giving A Higher Recommendation:**

The paper is clear, but not correct or reproducible. It doesn't meet the basic requirements of a "Tiny" paper; reproducibility requires far more details on the experimental setup; accepting 100% test data accuracy requires evidential clear explanation and justification.

**Reason For Not Giving A Lower Recommendation:**

N/A

---

### Decision · Program_Chairs · 2023-04-10

No revision received; not invited to archive